# The Influence of Q & T Heat Treatment on the Change of Tribological Properties of Powder Tool Steels ASP2017, ASP2055 and Their Comparison with Steel X153CrMoV12

**DOI:** 10.3390/ma17050974

**Published:** 2024-02-20

**Authors:** Jana Escherová, Michal Krbata, Marcel Kohutiar, Igor Barényi, Henrieta Chochlíková, Maroš Eckert, Milan Jus, Juraj Majerský, Róbert Janík, Petra Dubcová

**Affiliations:** 1Faculty of Special Technology, Alexander Dubcek University of Trenčín, Ku Kyselke 469, 911 06 Trenčín, Slovakia; jana.escherova@tnuni.sk (J.E.); michal.krbata@tnuni.sk (M.K.); igor.barenyi@tnuni.sk (I.B.); henrieta.chochlikova@tnuni.sk (H.C.); maros.eckert@tnuni.sk (M.E.); milan.jus@tnuni.sk (M.J.); juraj.majersky@tnuni.sk (J.M.); 2Faculty of Industrial Technologies in Púchov, Alexander Dubček University of Trenčín, Ivana Krasku 491/30, 020 01 Púchov, Slovakia; robert.janik@tnuni.sk (R.J.);

**Keywords:** tool steel, hardness, microstructure, ball-on-disc, coefficient of friction, wear

## Abstract

In connection with the growing importance of the efficiency and reliability of tools in industrial sectors, our research represents a key step in the effort to optimize production processes and increase their service life in real conditions. The study deals with the comparison of the tribological properties of three tool steels, two of which were produced by the powder metallurgy method—ASP2017 and ASP2055—and the last tool steel underwent the conventional production method—X153CrMoV12. The samples were mechanically machined with the finishing technology of turning and, finally, heat treated (Q + T). The study focused on the evaluation of hardness, resulting microstructure, wear resistance, and coefficient of friction (COF). The ball-on-disc method was chosen as part of the COF and wear resistance test. The tribological test took place at room temperature with dry friction to accelerate surface wear. The pressing material was a hardened steel ball G40 (DIN 100Cr6). Measurements were performed at loads of 10 N, 6 N, and 2 N and turning radii of 13 mm, 18 mm, and 23 mm, which represents a peripheral speed of 0.34, 0.47, and 0.60 m/s. The duration of the measurement for each sample was 20 min. The results showed that the COF of powder steels showed almost the same values, while a significant difference occurred with the increase of the radius rotation in the case of conventional steel. The results within the friction mechanism showed two types of wear, namely, adhesive and abrasive wear, depending on the Q + T process. From a tribological point of view in terms of wear, it was possible to state that the material ASP2055 after Q + T showed the lowest rate of wear of all the tested steels.

## 1. Introduction

Surface integrity is an important factor for components subject to wear, such as cold cutting tools, which must have high hardness combined with high wear resistance [1,2]. Various finishing technologies, such as grinding, hard turning, and hard turning together with polishing, have been developed in the industry to improve it. Heat treatment (HT) plays a key role and is usually carried out in vacuum furnaces with gas hardening to the required hardness [3]. COF and wear are key concepts in structural materials and engineering [4]. They play an important role in the analysis and design of mechanical systems and are at the center of attention because they reduce the efficiency and durability of materials. Wear, on the other hand, refers to the process of gradual removal of material from a surface due to friction and abrasion, which has a significant negative impact on the durability and performance of materials as it can lead to their degradation and failure. Many authors in the field of tribological properties focus on different materials and changes in input parameters. The main attention is paid to tool steels, where increased emphasis is placed on their functionality and durability in friction properties research. The research team led by Marek Hawryluk et al. [5] carried out an assessment of the influence type of steel in terms of tool life using the ball-on-disc method. The results indicated a clear presence of abrasion wear and also signs of material bonding in these locations. The authors stated that a higher tendency to plastic deformation is related to a lower occurrence of oxides on the steel surface. They identified the importance of chromium content in steel, where a higher concentration can act as protection against oxidation. Tobola and Łętocha [6] also investigated Vancron and Vanadis tool steels with different hardnesses. The results showed that Vanadis had a significantly higher wear rate, which was attributed to the presence of harder MC carbides. The authors declared that adhesive wear was the main dominant wear mechanism. Authors Kováčiková et al. [7] investigated tool steels in the context of dry friction with different bearing balls. The goal was to analyze the wear of steel and compare different loads. They found that increasing the load led to significant degradation of the sample surface. The results showed that the carbide content of the steel had a significant effect on friction and wear, with adhesive wear being the dominant mechanism and the hardness of the materials affecting the wear rate. Krbata et al. [8] also investigated the tool steel using a dry Si_3_N_4_ ball. The goal was to analyze the friction properties and compare the wear of steel depending on the friction speed. The authors found that an increase in the friction speed led to significant degradation of the material on the surface of the sample; also, in this research, the effect of hardness on wear was also demonstrated.

Changing the method of steel production from conventional to powder metallurgy (P/M) allows for increased tool fatigue strength and obtains steels that have a uniform fine grain structure characterized by higher hardness and toughness. The use of a suitable combination of carbides in the structure of tool material also affects the properties [9].

Young Keun Park et al. [10] investigated AISI M2 high-speed tool steel under different conditions. The authors’ wear results revealed that there was almost no damage to the M2 alloy when the bearing steel ball was used. When a ZrO_2_ ball was used, a measurable but small weight loss due to M2 wear was observed. Through their research, the authors concluded that wear tests using ZrO_2_ balls revealed that the wear rate of the M2 alloy depends significantly on the wear load but not significantly on the wear sliding speed. Zbynek Studeny et al. [11] also focused on tool steels of type M390 and M398, which are produced by powder metallurgy–HIP methods. The research of these authors showed that the new M398 material can fully replace the M390 material because it exhibits significantly better tribological properties. The M398 material showed more than 400% reduction in wear compared to the M390 material. The ideal HT consisted of a cryogenic quench to −78 °C and a tempering temperature of 400 °C. Faruk Çavdar [12] investigated the friction and wear properties of AISI D3 material. Its objective was to investigate the variation of COF wear track depth and width, volume loss, and wear rate of WR under different experimental conditions. The author concluded that ANOVA analysis revealed that feed distance had no significant effect on COF, indicating the uniformity of the wear testing specimen. The highest COF was observed at a load of 1 N and a speed of 100 mm/s. Low COFs were found at loads of 1 and 20 N at high speeds and 12.5 N at low speeds. An increase in speed resulted in an increase in track/groove depth at low distance values, while higher distance values resulted in a decrease in track depth with speed. The author noted that track width generally increases with extreme values of speed and load across all values of distance. Author Daniel Toboła [13] studied phase transformations in the surface layer of Vanadis 6 tool steel obtained after several processing methods. The author investigated changes in wear resistance. The samples were subjected to HT in gas-cooled vacuum ovens. In his study, the author performed tribological tests using the ball-on-disc method using the counterpart Al_2_O_3_. Wear resistance decreases while the wear rate increases with applied force, as in hard turning. E. Huttunen-Saarivirta et al. [14] investigated the tribological pair of H13 alloy steel and aluminum alloy at room temperature and elevated temperatures. The authors state that with increasing pressure, the wear mechanism also changes from abrasive wear at 510 MPa to adhesive wear at 810 MPa. The authors concluded that oxidation significantly increases tool steel wear, leading to greater material loss as the test temperature increases. At the end of the experiment, the authors stated that the COF and wear of the tribocouple samples depended on the applied contact pressure and the prevailing temperature. Määttä et al. [15] found that the composition of the tool steel does not have a marked effect on the friction between the tool and the workpiece. However, the surface roughness and topography of the tool have a significant influence. A study by Ceschini L. et al. [16] noted that the composition and sintering conditions had a considerable effect on the resistance of powder steels to sliding wear. In this regard, the best behavior was noted for more hardenable Fe-C-Mo steels with a higher Mo content (1.5% *w*/*v*), sintered under conditions resulting in the formation of a bainite microstructure. The authors’ results showed that the heating temperature and the cooling medium affect the wear characteristics of mild steel. The authors concluded that the high heating temperature and the rapid cooling medium, which was water, improved the mechanical properties of the mild steel. Authors Slawomir Swirad et al. [17], in their research, focused on polishing the surface of balls using the ball-on-disc method. Tests revealed that stroking the balls in most cases resulted in minimization of friction and wear of the contact elements. In most cases, samples characterized by high wear also resulted in high COF. The authors concluded that due to ball polishing, volumetric wear was reduced by up to 2.2 times, while COF was reduced by 1.7 times. Qi Wen et al. [18] experimentally investigated the COF using the ball-on-disc method with a rough surface in dry sliding contact. The authors’ results showed that the COF first increased and then became stable, in which the ramp-up and steady-state periods were included. The authors reported that as the normal load and rotational speed increase or the surface roughness decreases, the run-in duration and variation almost decrease.

This paper focuses on the research of three tool steels that have been machined by the turning process. Subsequently, the samples went through the final heat treatment process—quenching and tempering (Q + T). The presented study of this contribution focuses primarily on the investigation of the influence of HT on the resulting tribological values, which will mainly focus on the comparison of the wear rate, the overall wear mechanism, and the evaluation of the COF. The secondary goal of the presented contribution will be the evaluation of the change in hardness after HT and the comparison of the roughness of the grooves formed after the tribological process. There will also be verification of the chemical composition of the matrix materials as well as their carbide particles using EDS analysis. Experimental research will contribute to the scientific knowledge of how the heat treatment of the investigated tool steels affects their resulting wear and can provide a comprehensive view of the resulting wear mechanics in the tribological process.

## 2. Materials and Methods

Two tool steels, ASP2017 and ASP2055, which were produced by the powder metallurgy method, and one conventional tool steel, X153CrMoV12, were used for the experiment. We include ASP2017 and ASP2055 steels among high-speed steels that are produced by the powder metallurgy process [19]. ASP2017 steel is characterized by high toughness and excellent grindability. It is particularly suitable for producing taps or bimetal saws, while ASP2055 steel is characterized by a very fine carbide structure and is highly alloyed. These types of steel are suitable for producing forming, face mills, drawing mandrels, and various powerful tools for cold work. The last investigated steel was tool steel X153CrMoV12. It is a chrome–molybdenum–vanadium steel, which is characterized by excellent hardenability and good toughness. It is suitable for highly stressed cutting tools and tools for hot forming with high requirements for hardness and wear resistance when hot. Image interpretations found in Figure 1 represent the microstructures of ASP2017, ASP2055, and X153CrMoV12 steels, which were made using a scanning electron microscope (SEM) TESCAN VEGA 3 (TESCAN GROUP, Inc., Kohoutovice, Czech Republic) to compare the microstructure.

To verify the chemical composition of the experimental materials ASP2017, ASP2055, and X153CrMoV12, a spectral analysis of the chemical composition was also performed using a SPECTROMAXx LMX10 device (SPECTRO Analytical Instruments GmbH, Kleve, Germany). As part of the spectral analysis measurement, five measurements were performed for each experimental steel, from which the average value was calculated as indicated in Table 1. In all three cases, the measured chemical composition corresponds to the regulation given by the material sheet.

All samples were supplied in the basic soft annealed state. They were processed using the finishing technology of turning and then finally heat treated (Q + T). Heat treatment of steel consisted of quenching with austenitization at a temperature of 1070 °C in a vacuum, followed by rapid cooling in water. Heating to the austenitization temperature was two-stage with a short duration at 600 °C and then 850 °C. The two-stage heating served to reduce the possibility of cracking during the phase transformation of ferrite to austenite. Subsequently, the samples were tempered three times to a temperature of 550 °C (ASP2017 and ASP2055), respectively 510 °C (X153CrMoV12). Multi-stage tempering ensured the exclusion of secondary carbides.

The illustrations (Figure 1a–c) interpret the microstructures of ASP2017, ASP2055, and X153CrMoV12 steels, which were supplied in the basic soft annealed state. The structure in Figure 1a is formed by larger primary carbides and smaller secondary carbides that were formed during annealing. The carbides are regularly distributed in the ferritic matrix. The structure shown in Figure 1b also defines steel with uniformly distributed carbides in the ferritic matrix, which occur in increased quantities since this material contains a larger amount of carbide-forming elements based on Cr, W, V, Mo, as well as Nb. The third type of structure (Figure 1c) indicates the basic microstructure of X153CrMoV12 steel, which in the basic state was formed by a ferritic matrix with the occurrence of large primary carbides M_7_C_3_ and small secondary carbides M_23_C_6_. Secondary carbides were formed as a result of precipitation during steel annealing. The remaining illustrations (Figure 1d–f) represent the microstructures after the final heat treatment (Q + T). The structure in Figure 1d, steel ASP2017, is formed by larger primary carbides based on Mo and W and smaller secondary carbides based on Cr, which were formed during the tempering process. Nb-based carbides are dimensionally larger than ASP2017 steel [20]. Carbides are regularly distributed in the matrix of tempered martensite. The microstructure of ASP2055 steel (Figure 1e) is very similar to ASP2017 steel, also with uniformly distributed carbides in the matrix of tempered martensite, with the difference that these carbides reach larger dimensions. The structure (Figure 1f) of X153CrMoV12 steel is formed by prominent primary carbides and smaller secondary carbides. Secondary carbides were formed as a result of precipitation during the tempering of steel. Carbides are also regularly distributed in the matrix of tempered martensite.

For a more detailed study of the structure of all experimental steels, a chemical analysis was also carried out in the presented studies by means of EDS on a Tescan Vega 3 electron microscope, which was localized only to specific chemical elements [21,22]. The measurements were focused mainly on carbide particles, and thus, the results do not correspond to the results of the spectral analysis, as carbides have a different chemical composition and lattice. Figure 2, Figure 3 and Figure 4 interpret the distribution of individual chemical elements for all investigated steels through EDS analysis, which is expressed by the color scale [23].

The resulting values of the chemical composition of these steels through EDS analysis are clearly documented in Table 2. The EDS analysis of ASP2017 steel shows that large primary carbides are mainly composed of W and Mo. Fe is mainly concentrated in the matrix. For ASP2055 steel, the results of the analysis were similar to the previous case. This steel contains a larger amount of Nb, which is more pronounced in the composition of primary carbides [20]. In the chemical composition of the steel X153CrMoV12, the main alloy C dominates, the occurrence of which is mainly concentrated in large primary carbides M_7_C_3_ [24]. Fe is again mainly concentrated in the matrix. Other elements are represented in a minority, and their color distribution within the EDS analysis is difficult to see.

As part of the experiment, hardness measurements of all investigated materials ASP2017, ASP2055, and X153CrMoV12 were also made, performed on samples in the basic state, and turned Q + T samples. The hardness of the selected samples was measured at five independent locations, from which the average value for a specific type of test sample was calculated. As part of the hardness measurement, the Qpix T2 software version 3.1.0.4 was used, which was part of the equipment of the QATM hardness tester. As part of hardness testing, the Vickers-type method was applied with a load of 10 kg (HV10). The standard STN EN ISO 6507-1 (420374) [20,25] was used for hardness testing.

The tribological experiment was carried out on a UMT TriboLab tribological tester from Bruker. As part of the experimental testing, the tribological method of the ball-on-disc type was applied, which is one of the most used methods in the field of tribology, serving not only to measure COF but very often it is also possible to encounter it when evaluating the resistance of individual materials to wear [26,27,28]. The method used is suitable for investigating the properties of various materials, including metals, polymers, ceramics, composites, and other types of materials. A sample of the experimental sample after tribological testing and its dimensions are clearly shown in Figure 5. A hardened steel ball made of material G40 with a diameter of 4.76 mm was chosen as the counterpart contact. Subsequently, three types of loads were chosen—10 N, 6 N, and 2 N, as well as three types of turning radii—13 mm, 18 mm, and 23 mm, which represent a peripheral speed of 0.34, 0.47, and 0.60 m/s. All tribological measurements were carried out according to the valid ASTM G99 standard [29] at room temperature without the application of lubricating liquid to accelerate the degradation of the material. The measurement duration for each experimental sample was 20 min. The rotation speed was set to 250 rpm. COF values and wear of individual types of materials were measured and evaluated in relation to HT (Q + T), applied load, and turning radius.

When analyzing the resulting wear of the surfaces of the test materials and the counterpart, a laser scanning confocal microscope LSCM Olympus LEXT OLS5100 was applied, where the 3D roughness of the selected surfaces of the test materials was also made.

## 3. Results and Discussion

### 3.1. Hardness of Materials

From the graphic interpretation (Figure 6), it is clear that, among all the examined steels, the ASP2055 sample achieved the highest hardness in both cases of processed sample forms (Base samples—Turned samples Q + T), which reached a value of 910 HV10. With the ASP2017 material, which is produced using the same powder metallurgy technology, the lowest hardness of 842 HV10 was achieved after double tempering. With the last third material, X153CrMoV12, a hardness of 895 HV10 was achieved. The hardness was measured at five places on each sample, from which the average value for a specific type of test sample was subsequently calculated. The Qpix T2 software (version 3.1.0.4) was used to measure the hardness. The hardness results point to the fact that the influence of alloying elements plays an important role in the final hardness values of steels [30]. The ASP2055 material contains a high amount of alloying elements that support the formation of carbide particles based on Cr, W, V, Mo, and Nb, and the steel itself also contains a higher percentage of C, which again supports the formation of these hard particles. This chemical composition ultimately influenced the so-called highest hardness value [15,31]. The hardness values of the V, Nb, and W-based carbides themselves range from 2700 to 2500 HV0.05 [32,33,34]. The ASP2017 material is characterized by the same type of alloying elements, with the difference that the resulting ratio is reduced by more than half, mainly in the content of C, V, Nb, and W. For this reason, this material reached the lowest values of the final hardness. Conventionally produced tool steel X153CrMoV12 achieved almost the same hardness value as ASP2055 steel due to the high Cr content, which formed primary carbides M_7_C_3_ and small secondary carbides M_23_C_6_. These carbides reach a hardness in the range of 900–1700 HV0.05 [35].

### 3.2. Coefficient of Friction

An example of the comparison shown in Figure 7 shows the measured COF curves for the material X153CrMoV12, turning radius 23 mm, load 10 N. Two zones can be observed in this illustration. Zone A represents the start-up part of the curve, where a friction groove begins to be generated on the surface of the examined material within 180 s. Zone B already defines the steady part of the curve. It is also possible to see that the highest COF value within the “Ball-on-Disc” method was achieved at the interface of both zones for both types of samples. Subsequently, the COF value stabilized and began to show regular fluctuations in values. This oscillation of values is caused by the accumulation and release of these microparticles of material in front of the counterpart. This material creates resistance during friction, which is subsequently released, and the COF value decreases. During this process, two ways of its movement occur: the first part of the material moves to the edge parts of the friction track, which is also affected by the centrifugal force during the experimental testing, and the second part of the material is pushed into the surface of the friction track by a counterpart in the form of a ball from which later become abrasive particles. These particles are exposed to a large plastic deformation during the friction process, which also increases their hardness and strength, which affects the increase in the wear value of the material [36,37].

Figure 8 defines the comparison of COF depending on the turning radius of individual types of materials. In this illustration, only the 10 N load measurements were compared to each other because no significant changes in the COF values were observed at the lower 6 N and 2 N loads. A comparison of the COF values for the ASP2017 material can be observed in Figure 8a. From the graphical interpretation, it can be concluded that as a result of Q + T, there was a significant increase in the COF values at all three turning radii. The lowest COF value was obtained for the non-heat-treated sample at a radius of 23 mm, with a COF value of 0.735. The highest value achieved in this comparison was 0.937 on a sample that underwent the Q + T process with a turning radius of 13 mm. From an overall point of view, it can be concluded that COF values increased by ~ 20% during the Q + T process. Figure 8b shows a comparison of the COF values from the radius of gyration for the tested material ASP2055. The graphical representation defines again the difference between the heat-treated and the untreated sample, as was the case with the previous ASP2017 material. From an overall point of view, it could be assessed that there was a re-increase in COF values at Q + T by ~18%, which could be considered a similar result to the ASP2017 material, as it is also a PM steel with the same alloying elements.

In the case of the last experimental material, X153CrMoV12 (Figure 8c), a different situation occurred than in the case of the previous experimental materials. The basic sample, with a turning radius of 13 mm, showed a higher COF value compared to the sample that went through the Q + T process. The COF value on the given sample was 0.78, while on the Q + T sample, it reached a value of 0.73. The achieved COF value on the basic sample for radii of 23 mm was 0.59, which represents a ~ 50% drop in COF compared to the sample with Q + T.

The overall COF results point to the fact that the steels that underwent the PM production method show similar COF values with respect to the radii of rotation with the set tribological parameters and their fine-grained microstructure. The final heat treatment leads to a significant increase in hardness, which is closely related to an increase in COF values [38]. The results of the conventional steel X153CrMoV12 in the case of the base material led to a decrease in COF due to an increase in the friction temperature, which is caused by an increase in the turning radius [39]. The reasons for the increase in COF on samples that have undergone the Q + T process due to the friction mechanics analysis on these samples will be discussed in the sections below.

### 3.3. Wear

A comparison of the wear of all investigated steels for a turning radius of 13 mm, where there was a significant difference between the samples that underwent the Q + T process and the samples that were not subjected to Q + T, is shown in Figure 9. The resulting wear value is, therefore, cumulative, i.e., the volumes of removed and extruded material make up the total wear of the samples. The orange color on the graph indicates the volume of material that has been removed, while the blue color indicates the plastically deformed material that has pushed out the sides of the friction track. For samples without final Q + T, it was seen that the lowest wear value was achieved by the ASP2017 material, which also showed the lowest amount of extruded material on the sides of the friction groove. The highest value recorded was by the ASP2055 material. This material also showed the largest percentage difference in wear rate. Figure 10 shows a comparison of the wear of all the investigated steels for a turning radius of 18 mm, where a significant wear difference was observed between Q + T samples and base samples. Similar to the previous material, a similar ratio of wear values was observed, with the lowest wear observed on the ASP2055 material after Q + T. Of the samples that did not go through the Q + T process, the ASP2017 material again showed the least amount of wear.

In contrast to the previous two turning radii, different wear values were obtained for base samples for a turning radius of 23 mm (Figure 11). The ASP2017 material achieved the greatest wear in the given samples. Also, as in previous cases, this material showed only a minimal volume of printed material. The lowest wear was achieved by the material X153CrMoV12, which, when compared to the material ASP2017, achieved approximately 50% wear. The same result was achieved for the samples that went through the final Q + T process as for the other turning radii. From these results, the ASP2055 material again showed the lowest wear rate.

From an overall point of view, it can be concluded that Q + T played an important role in the wear process for all investigated steels. ASP2055 steel, which is produced by the PM process, achieved the lowest wear rate with respect to all three turning radii. The reason for the lowest rate of wear in the given material is the occurrence of a fine-grained microstructure, which consists of a very hard martensitic matrix and a large number of uniformly distributed globular carbide particles. Figure 12 compares the enlarged areas of wear volumes after Q + T of individual experimental tool steels. The results clearly show that ASP2055 steel achieved the lowest wear rate at all turning radii. It is also possible to observe that the ratio of removed and printed material reached a ratio of ~3:1. The results of wear were closely linked to the mechanics of its occurrence, respectively, the type of wear that occurred, which is evaluated in more detail in chapter 3.4 Wear Mechanisms.

### 3.4. Wear Mechanisms

The detailed determination of the friction mechanism and the determination of the occurrence of individual types of wear are shown in Figure 13. Due to the extensiveness of the research and the resulting data, only one turning radius, 23 mm, was chosen for evaluation, as there were no significant differences in the result.

In the case of samples with Q + T (Figure 13b,d,f), exclusively abrasive wear was detected due to their significant strength or hardness. The formation of abrasive wear was caused by the friction process itself, during which microabrasive particles were formed, which subsequently hardened plastically and thus increased their strength. These particles rolled between the experimental material and the pressure friction ball and formed microgrooves [40] on the surface of the friction groove, which occurred on all kinds of experimental materials after Q + T. On the surface of the friction surfaces of the heat-treated samples, there was a deposit of an oxidation layer, which was most detected on the ASP2055 material (Figure 13d). In the case of the other two heat-treated samples, it was detected to a much smaller extent. In the case of a comparison of the experimental materials ASP2055 and the pair of materials ASP2017 and X153CrMoV12, it was seen that in the pair of materials, the oxidation layer was more finely distributed, while in ASP2055, it formed a more complete layer. This integrated layer on the given material created the so-called solid lubricant that prevented direct contact between the ball and the base material and thus prevented its wear [41,42]. The occurrence of deep grooves was also observed with the given material.

When comparing base samples (Figure 13a,c,e), a different surface morphology was seen, which mainly showed only adhesive wear. Since these materials had significantly lower hardness, significant plastic deformation in the friction grooves could therefore occur. In this case too, the deposit of an oxidation layer was detected, but to a much greater extent. In Figure 13a, a mixed type of wear, i.e., abrasive–adhesive, was evaluated, while the abrasive wear was much smaller. During adhesive wear, there was a formation of microdeposits between the contact pair of materials (ball + experimental sample), which caused a constant change in the roughness of both surfaces and thus led to the occurrence of plastic deformation. The occurrence of plastic deformation was also caused by the state of hardness, while it is generally true that the proportion of plastic deformation in the friction process increases with decreasing hardness [43,44].

For a more detailed examination of the overall friction mechanism, G40 balls were also evaluated, which are shown in Figure 14. Figure 14a,c,e show the profiles of the balls in contact with experimental materials that did not undergo the Q + T process. In contrast, Figure 14b,d,f show the profiles of G40 material beads after contact with Q + T materials. It has been observed that in contact with ASP2017 steel, oxidative wear occurred on both types of ball surfaces, which is limited to small areas (Figure 14a,b). The occurrence of deep parallel grooves was also observed in the given materials without Q + T (Figure 14a). In contrast, in contact with the ASP2055 material, oxidative wear occurred to a much lower extent (Figure 14c,d). As in the previous case, a large number of shallower grooves appeared on the base material ASP2055 without Q + T. At the last surfaces of the G40 balls that moved over the X153CrMoV12 materials, a very fine oxidation layer occurred, which in the sample without Q + T occupies almost the entire contact area of the ball (Figure 14e). Overall, it can be concluded that the change in the heat treatment of the experimental materials led to a change in the mechanics of the contact surfaces of the G40 balls. On the surface of the balls that moved over the samples without Q + T, there was predominantly oxidative wear with a large occurrence of fine grooves. Whereas, on the surface of the balls that moved over the samples with Q + T, there was only a small amount of oxidative wear with a smaller number of deeper grooves.

### 3.5. Surface Roughness

Using LSCM Olympus LEXT OLS5100, 3D images of the surfaces of the test materials ASP2017, ASP2055, and X153CrMoV12 were made. Materials with a turning radius of 23 mm were chosen as representative samples (Figure 15). The displayed results pointed to the fact that samples after Q + T showed significantly lower wear values compared to base samples. In the given illustrations (Figure 15), it was also seen that all the samples underwent the same machining process (turning) and, therefore, achieved the same surface asperity before the start of the tribological measurements.

Figure 15a,c,e show the 3D groove profile without Q + T, where significant wear can be clearly observed within all types of experimental tool steels. There is also slightly detectable plastic deformation in the form of printed material along the edges of the friction grooves. On the test samples, when examining specimens ASP2017 and ASP2055 (Figure 15b,d), it was notably challenging to identify the friction groove in comparison to the remaining specimens. This observation strongly suggests that these materials exhibit an exceptionally low total wear rate following the Q + T process. On the X153CrMoV12 sample (Figure 15f), it was possible to optically detect a slightly higher wear value than in the previous samples. All these results were closely related to the wear comparison in Figure 11.

## 4. Conclusions

The article in question focused on the change of tribological properties of three tool steels, ASP2017, ASP2055, and X153CrMoV12, which were produced by the turning process and underwent the final heat treatment (Q + T) process. The whole experiment was carried out without the use of lubricant, with a load of 10N and three turning radii. A hardened steel ball of material G40 was used as the counterpart. The secondary goal of the presented contribution was the evaluation of the change in hardness after Q + T and the comparison of the roughness of the formed grooves after the tribological process. The chemical composition of the matrix materials, as well as their carbide particles, were also verified using EDS analysis. The following conclusions can be drawn from the presented work:
Heat treatment Q + T significantly affects the increase in hardness values. All experimental steels achieved approximately a 200% increase in hardness compared to the base material.The type and amount of alloying elements affect the total content of carbide particles, which also affects the resulting hardness. The uniform distribution and high content of different types of carbides led to the fact that the material ASP2055 achieved the highest hardness of all the experimental materials of up to 910 HV10. The high hardness was due to the high content of alloying carbide-forming elements, especially Cr, W, V, Mo, and Nb.The COF results point to the fact that the ASP2017 and ASP2055 steels, which underwent the PM production method, show similar COF values with regard to the set tribological parameters and their resulting heat treatment. Their heat treatment in the form of Q + T leads to an increase in COF values on average by ~20%.The lowest wear rate of 0.01 mm3 with respect to all three turning radii was achieved by ASP2055 steel, which underwent the Q + T process. The reason for the lowest rate of wear in the given material is the occurrence of a fine-grained microstructure, which consists of a very hard martensitic matrix and a large number of evenly distributed globular carbide particles, as well as the fact that a complete oxidation layer has occurred on the material. The ratio of removed and extruded material after Q + T reached ~ 3:1 for all types of experimental materials.Within the friction mechanism, two types of wear occurred, i.e., adhesive and abrasive wear. At the same time, adhesive wear prevailed on base samples. These samples also experienced a greater degree of plastic deformation as part of this wear. Abrasive wear was clearly detected in samples with Q + T. This type of wear was associated with a change in the microstructure of the material after Q + T when a very hard martensitic matrix and carbide particles formed in the material. This heterogeneous structure (martensite + carbides) showed a very high hardness, but at the same time, internal stress also occurred in the material, which caused the brittleness of the material structure. The combination of these factors caused the formation of abrasive particles that were created in the friction process and created a typical abrasive surface that was formed by microgrooves.On the surface of the G40 counterpart balls, which moved over the samples without Q + T, there was predominantly oxidative wear with a large occurrence of fine grooves. On the surface of the balls that moved over the samples with materials with Q + T, there was only a small amount of oxidative wear with a smaller number of deeper grooves.The basic roughness of all experimental samples was at the same level and did not play a major role in wear. The samples without Q + T showed a high degree of plastic deformation in the form of extruded material on the edges of the friction groove.Tool steels can successfully represent wear-resistant steels because their chemical composition positively supports mechanical properties.

These materials can be used, e.g., to produce screws for injection machines in the plastics industry. Increasing the life cycle of these screws can lead to significant economic savings. Experimental research will contribute to scientific knowledge of how the heat treatment of the investigated tool steels affects their resulting wear and can provide a comprehensive view of the resulting wear mechanics in the tribological process.

## Figures and Tables

**Figure 1 materials-17-00974-f001:**
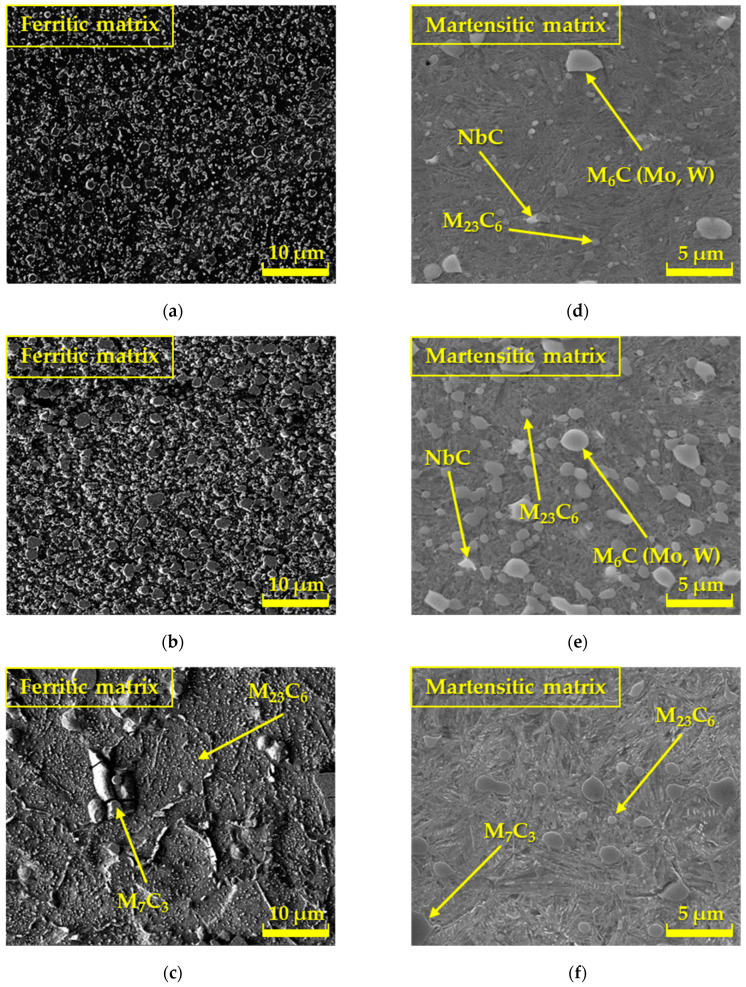
Microstructure of tool steels: SEM—basic condition—supplied in the soft annealed condition (**a**) ASP2017; (**b**) ASP2055; (**c**) X153CrMoV12; SEM—machined by finishing turning technology and heat treated by refining; (**d**) ASP2017; (**e**) ASP2055; (**f**) X153CrMoV12.

**Figure 2 materials-17-00974-f002:**
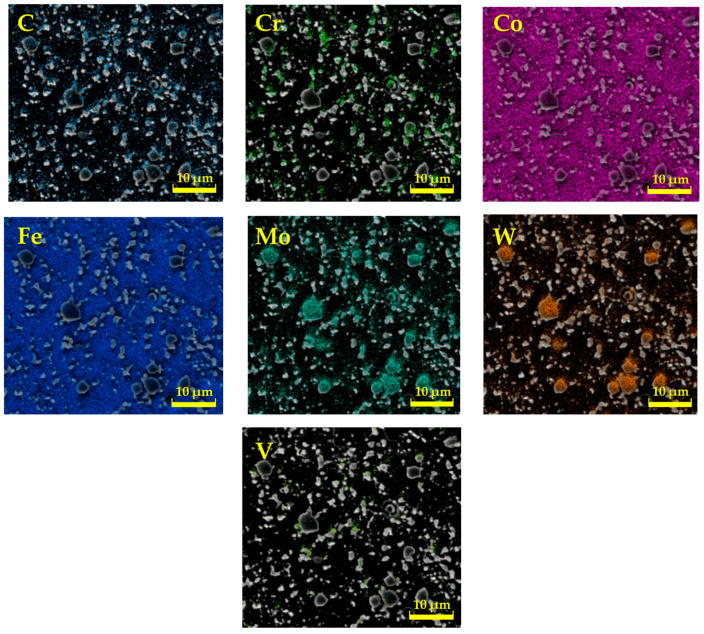
Analysis of the chemical elements of ASP2017 by EDS.

**Figure 3 materials-17-00974-f003:**
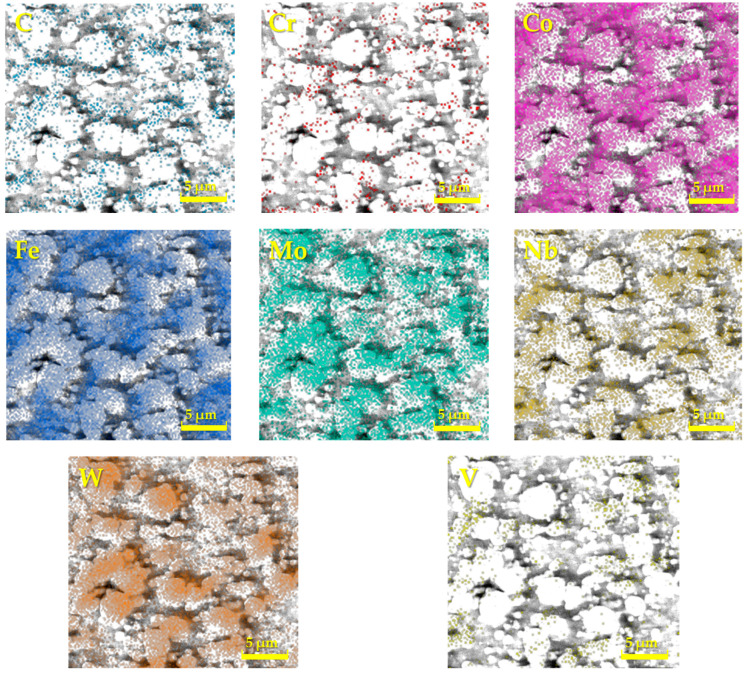
Analysis of the chemical elements of ASP2055 by EDS.

**Figure 4 materials-17-00974-f004:**
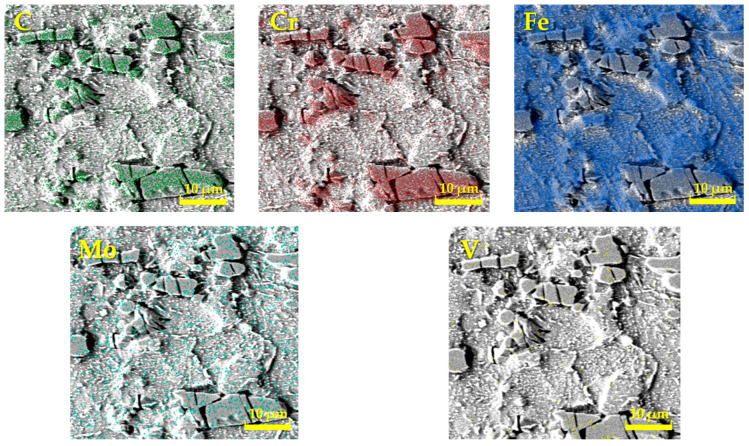
Analysis of the chemical elements of X153CrMoV12 by EDS.

**Figure 5 materials-17-00974-f005:**
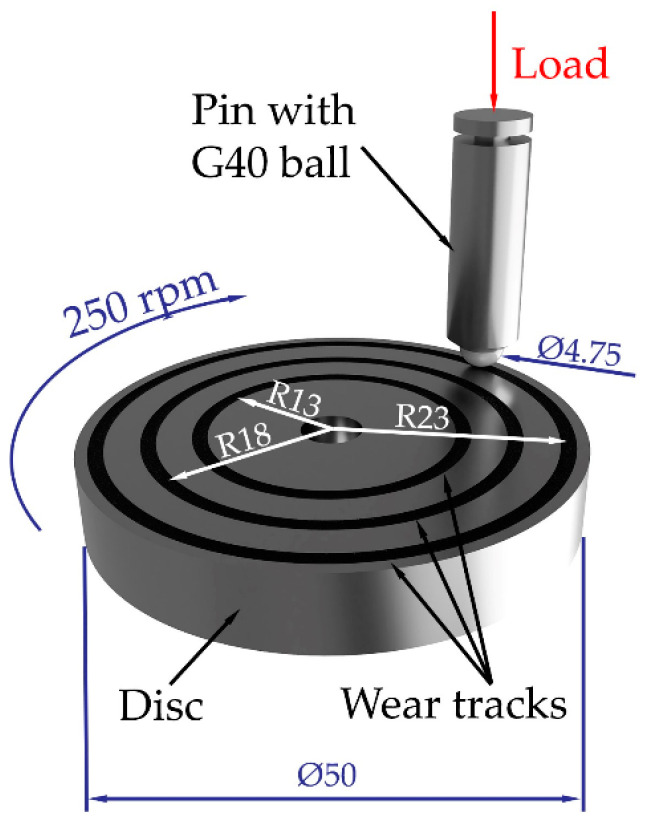
Example of an experimental sample after tribological testing.

**Figure 6 materials-17-00974-f006:**
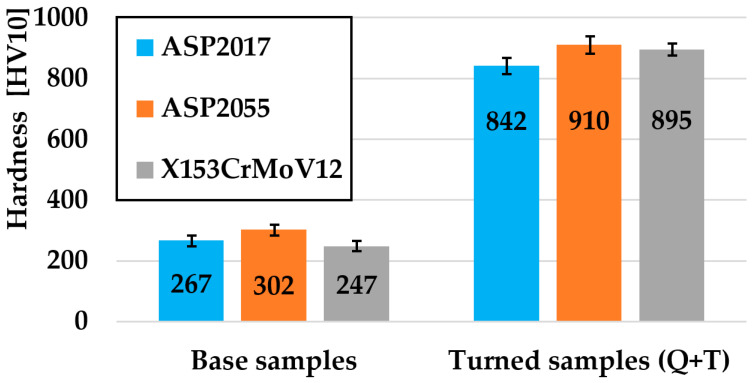
Graphic evaluation of hardness results for all investigated materials.

**Figure 7 materials-17-00974-f007:**
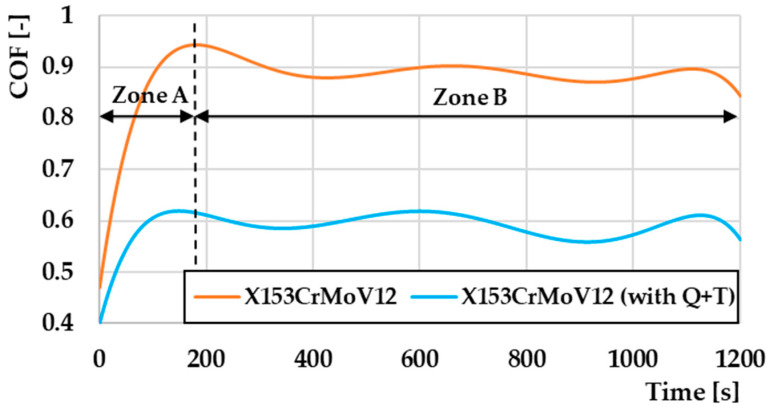
Comparison of COF values depending on the measurement methodology for material X153CrMoV12, turning radius 23 mm, load 10 N.

**Figure 8 materials-17-00974-f008:**
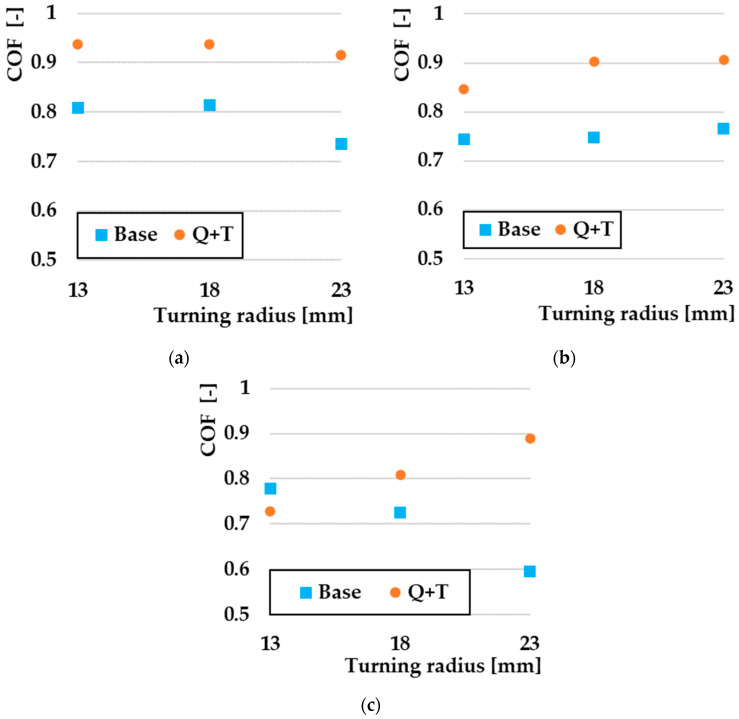
Comparison of COF by turning radius at 10 N: (**a**) ASP2017; (**b**) ASP2055; (**c**) X153CrMoV12.

**Figure 9 materials-17-00974-f009:**
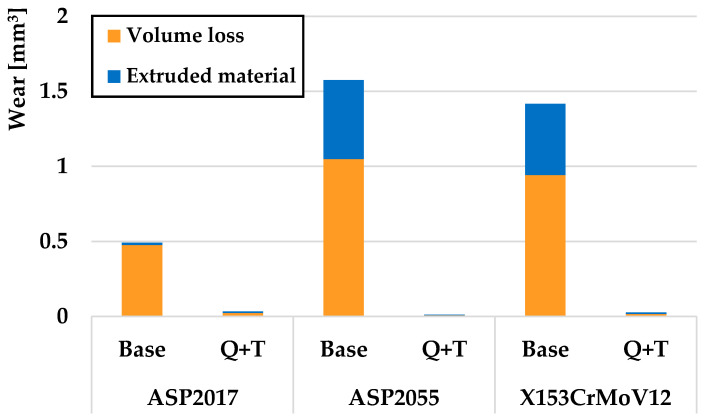
Volumes of removed and extruded material for a radius of 13 mm.

**Figure 10 materials-17-00974-f010:**
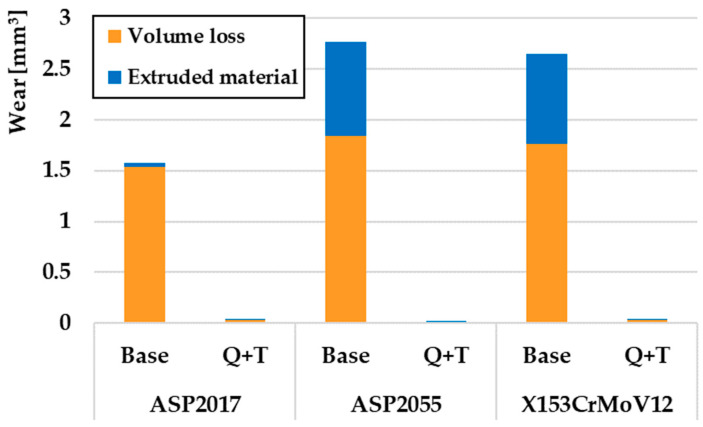
Volumes of removed and extruded material for a radius of 18 mm.

**Figure 11 materials-17-00974-f011:**
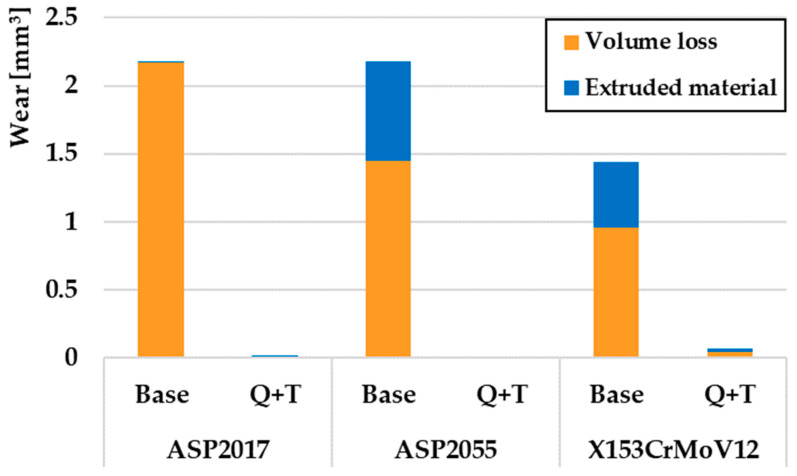
Volumes of removed and extruded material for a radius of 23 mm.

**Figure 12 materials-17-00974-f012:**
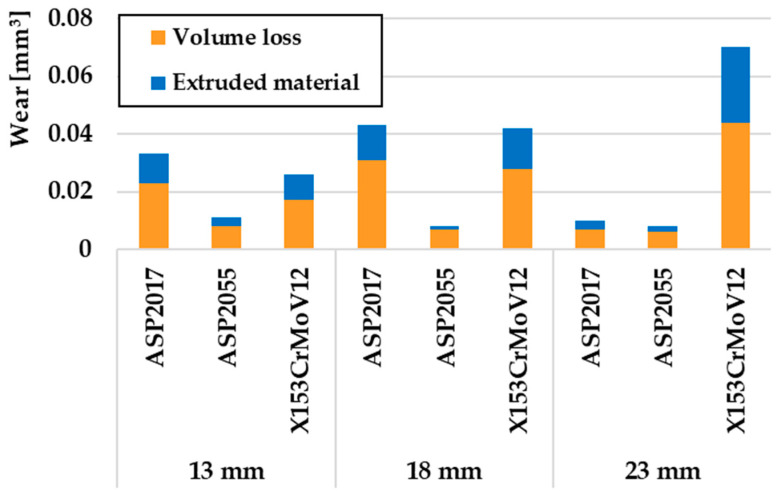
Comparison of volumes of removed and extruded material after the Q + T process.

**Figure 13 materials-17-00974-f013:**
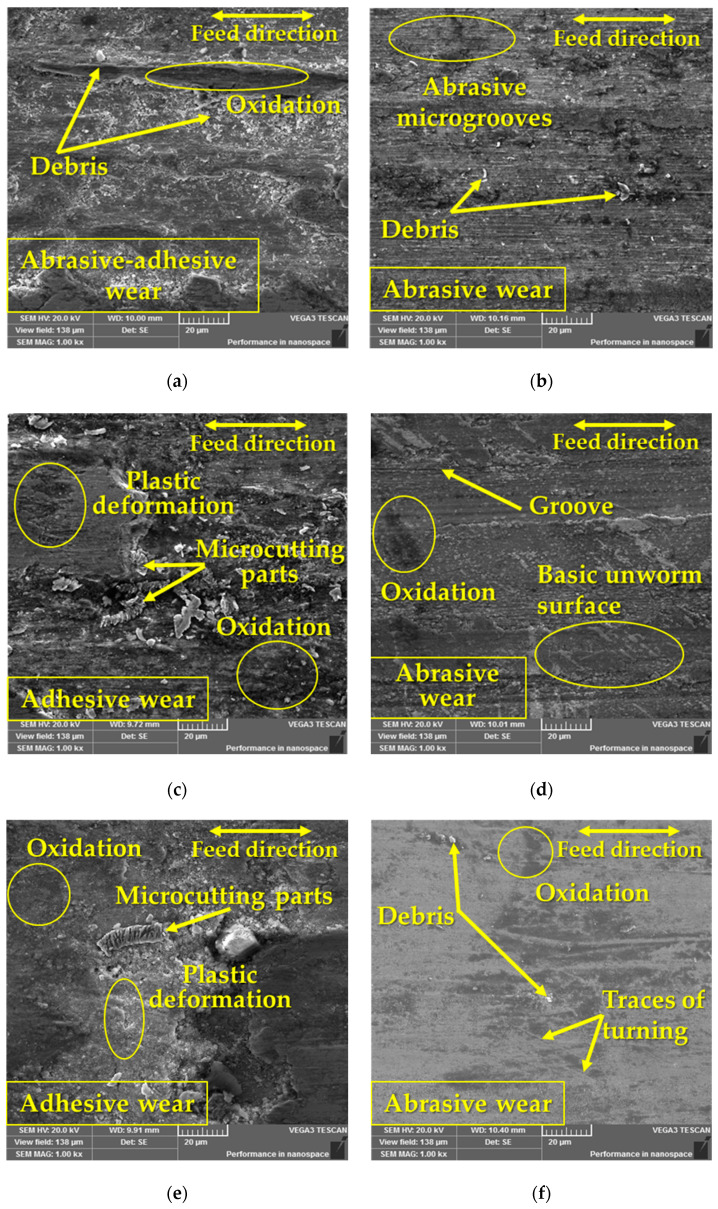
Wear analysis of the surfaces of the investigated materials at a radius of 23 mm: (**a**) ASP2017 base material; (**b**) ASP2017 with Q + T; (**c**) ASP2055 base material; (**d**) ASP2055 with Q + T; (**e**) X153CrMoV12 base material; (**f**) X153CrMoV12 with Q + T.

**Figure 14 materials-17-00974-f014:**
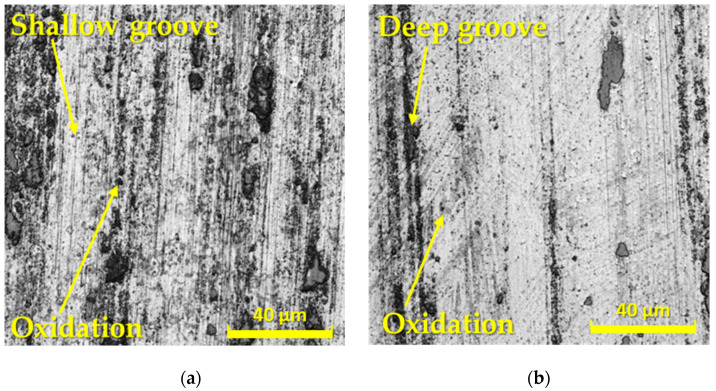
Wear analysis of G40 counterpart surfaces at a radius of 23 mm after contact with (**a**) ASP2017 base material; (**b**) ASP2017 with Q + T; (**c**) ASP2055 base material; (**d**) ASP2055 with Q + T; (**e**) X153CrMoV12 base material; (**f**) X153CrMoV12 with Q + T.

**Figure 15 materials-17-00974-f015:**
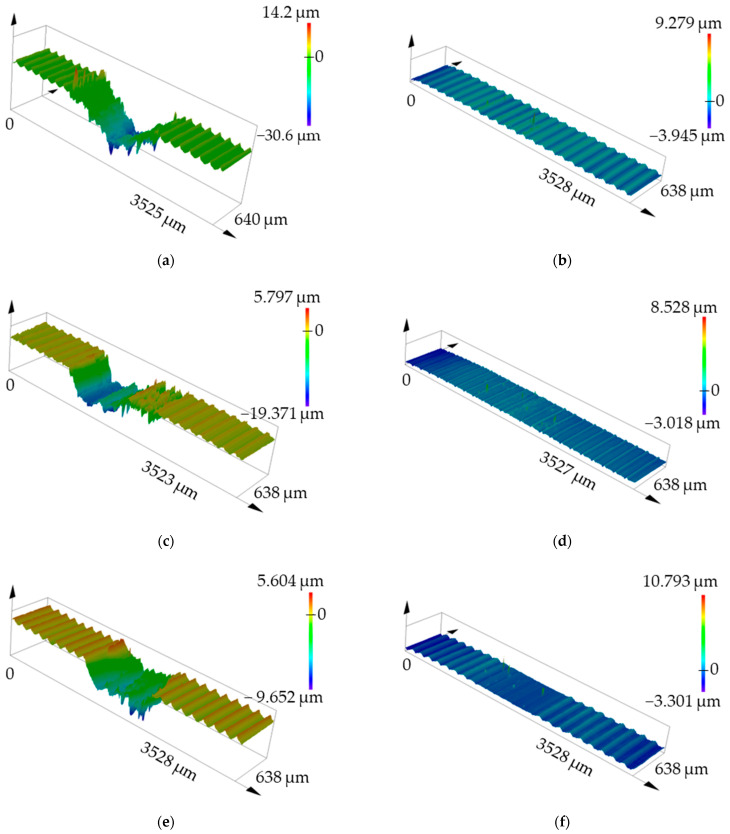
3D selected views of material surfaces at a radius of 23 mm. (**a**) ASP2017 base material; (**b**) ASP2017 with Q + T; (**c**) ASP2055 base material; (**d**) ASP2055 with Q + T; (**e**) X153CrMoV12 base material; (**f**) X153CrMoV12 with Q + T.

**Table 1 materials-17-00974-t001:** Chemical composition of the examined steels through spectral analysis.

Element [wt.%]	C	Cr	Mo	W	Co	V	Nb	Fe
ASP2017	0.76	3.91	3.13	2.70	9.07	0.99	1.26	78.18
ASP2055	1.63	3.92	4.55	5.55	9.88	2.53	2.59	69.35
X153CrMoV12	1.58	11.9	0.79	0.40	0.51	0.73	-	84.09

**Table 2 materials-17-00974-t002:** Chemical composition of the examined steels through EDS analysis.

Element [wt.%]	C	Cr	Mo	W	Co	V	Nb	Fe
ASP2017	4.75	7.94	4.63	5.96	1.75	2.55	4.51	72.42
ASP2055	7.62	10.79	7.18	14.27	3.00	7.65	4.51	44.99
X153CrMoV12	5.79	24.71	2.79	-	-	1.81	-	64.89

## Data Availability

Data are contained within the article.

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
