# Peer review of "The Influence of Q & T Heat Treatment on the Change of Tribological Properties of Powder Tool Steels ASP2017, ASP2055 and Their Comparison with Steel X153CrMoV12"

_materials, 2024, doi:10.3390/ma17050974_

Round 1
Reviewer 1 Report
Comments and Suggestions for Authors
The paper focused on the application of quenching and tempering heat treatment on three different steels (ASP2017, ASP2055 and X153CrMoV12). The work is experimentally complete, focusing on different tests, and the results are significant. The paper fits the scope of the Journal. However, there are significant issues to address, which are pointed out in detail in the attached PDF file.

Issues pointed out in the PDF file. Please do not speak in the 1st person.
Reviewer 2 Report
Comments and Suggestions for Authors
In this paper "The influence of Q&T heat treatment on the change of tribological properties of powder tool steels ASP2017, ASP2055 and their comparison with steel X153CrMoV12" the authors presented
comparison of the tribological properties of three tool steels, two of which were produced by the powder metallurgy method - ASP2017, ASP2055, and the last tool steel underwent the conventional production method - X153CrMoV12. The samples were mechanically machined with the finishing technology of turning and finally heat treated consisted of quenching with austenitization at a temperature of 1070 °C in a vacuum followed by rapid cooling in water. In the research, the authors focused on the evaluation of hardness, resulting microstructure, wear resistance and coefficient of friction (COF). To determine coefficient of friction and wear resistance, the authors used the ball on disc method.
Each of the presented parts of the publication has been correctly described by the authors. The introduction is correctly written and supported by very recent literature.The conclusions are consistent and closely related to the research topic.
Comments for authors:
1. I wonder why the authors used the turning radii for the analysis and not the friction path.
2. Figure 4. - What color was used to show Mo and V? I can't see it in the photo.
3. Figure 15. - Please enlarge the legend because you can't see anything.
In my opinion, the authors wrote and presented their research very well. The publications were a good read and I found answers to every question I had.
Reviewer 3 Report
Comments and Suggestions for Authors
Dear Authors, thank You for presenting this interesting manuscript. Here below are some remarks and suggestions, that may improve it.
Line 12-14. Please, give some arguments, why did You select these 3 different materials to study (maybe, they are used for identical applications?)? Explain, what is that "conventional method"? Is it casting?
Line 17. What are the other methods You use to study wear and CoF?
Line 18: dry friction instead of "without lubrication"
Line 20. Friction (wear) test - it is not the measurement. Please, use the correct terms. You can measure the distance or weight. But, it is not possible to measure the wear resistance.
The results showed that the COF of powder steels showed 21 almost the same values, while a significant difference occurred with the increase of the radius in the 22 cases of conventional steel. - This sentence is unclear. Do You increase the turning radius during machining, or do You mean the rotational (placement) radius of the ball on the disc?
Within the friction mechanism, there were two types of wear, adhesive and abrasive, which were affected by the Q+T process... Please, rewrite this sentence. Do You mean, that mean types of wear were adhesive and abrasive wear?
Line 33. "In practice" - what do You mean? In industry?
Line 34. Gas hardening? What do You mean? Nitriding, or gas carburizing?
Line 41-42. Input and output parameters: what do You mean? Please, explain
Please, explain what is steel class. Use internationally accepted terminology
The literature review is chaotic. There is no systematization, no general idea. Each next link leads to a completely different property or statement, not anyhow connected with the previous one. It can't be so. Also, You often mention "a tool steel" or "tested", but neither the chemical composition nor test conditions are not explained. Please, insert a "red line" in Your literature review.
Line 65 ... high-temperature wear tests.... How does this refer to Your topic?
Line 92. What is the pressure material? Counterpart?
Line 93: in this case, "wear rate increases" is preferable
Line 123-124: turning - is not a steel production method. Please, correct here
The literature review should end with a bullet statement that forces You to research the material. In Your case, the literature review lives its own life. There is no logical connection between the introduction and lines 123-134 (the aim and the task of the research). Also, the literature review should contain few links, presenting the results on exactly the same (or very-very similar) materials obtained before by other authors.
Line 136. Once again, the logic of selecting these 3 steels is not explained. Please, present the arguments.
Please, explain, how did You select the quenching and tempering temperatures. Give the references (Lines 141-145)
Another table should be placed in section 2 just after the studied materials are mentioned for the first time. It should contain the nominal composition of the steels.
Table 1. Please, give the Fe content in digits, as it is in Table 2.
So, You have 2 different tables, with completely dissimilar results. Which one is correct? How can we assess the adequacy of the tables, if we do not know the nominal composition of the steels?
Figure 1. As I think, You should do the additional test (EBSD, or micro-Xray diffraction) to prove, that exactly that kinds of carbides are formed. On the SEM pictures, there is no difference between NbC and M23C6. How they were distinguished?
Figure 1 and the supporting text should be moved to the "Results" section.
How were the samples for the SEM where prepared?
Give the correct (as it is required by the Editorial Board) descriptions of the used equipment
Lines 230-235. Please, add the sliding speed and the distance passed
Increasing the turning radius increases the sliding speed and test distance (the rpm and test duration, according to Your description, do not change). Thus, it causes the change of 2 basic parameters of the friction process. But, You do not mention this. I think, that the experiment should be done with only one parameter changing: sliding speed or sliding distance, but not both
This increase in the turning radius increases or decreases some resultant values. But, what are their individual inputs?
What is the need to test untreated (as-supplied) materials? It only overloads the manuscript and proposes unnecessary results. These tool steels are never used in this condition. Please, remove the friction test data for the as-supplied materials. Compare the results for 3 heat-treated steels only.
Combine Figures 9, 10, and 11 into one picture, the same as You did in Fig. 12
The presented light microscopy images can't be used to evaluate the wear mechanism: their magnification is too low, and most of the mentioned by the Authors details (like deep groove, shallow groove, and oxidation) can't be clearly seen on them. Much higher magnification and EPMA/EDS are also required. Please, do not waste Your time: remove the data for untreated material. Do the SEM images, as You did them in Fig. 1, and then - discuss about wear mechanism.
What is a "pressure friction ball"? Please, show it on the SEM image.
Please, add the full description of all used equipment and research methods in section 3. This will help to evaluate all the research methodologies and to catch the logic of the research. Do not start each new section/subsection with a description of another one instrument.
Line 399. Do not compare the treated material with untreated. Compare only the 3 heat-treated materials between themselves. And, with other results known from the literature.
Line 435-436: Materials with a turning radius of 23 mm were chosen as representative samples (Fig. 15)... Why?
Figure 15. The digits on the figure are too small, please, increase them
Add the "discussion" section, compare Your results with other known results, and make conclusions based on this comparison.
Line 465: ...steels achieved more than 200% increase in hardness How many? 700%? Please, specify here.
For all the research stages, please, compare Your results with the other like materials. As I read Your manuscript (with zero comparisons) I can't find, are these results good or not.

English style and grammar of the abstract should be extensively improved by a professional editor. Language issues alone are enough to reject the publication of the manuscript.
The manuscript itself is written OK, moderate editing is required.
Round 2
Reviewer 1 Report
Comments and Suggestions for Authors
A detailed report highlighting all changes made to the original manuscript should be provided
Comments on the Quality of English LanguageA detailed report highlighting all changes made to the original manuscript should be provided
Reviewer 3 Report
Comments and Suggestions for Authors
Dear Authors, despite You making some changes, it did not improve the manuscript.
Lines 12-14. It is even less understandable than it was before. Once again. What is "in practice"?
Line 19. The part of CoF measurement was done with BoD test configuration. And the rest of the measurements... What was used for them? Please, use comprehensive language. I don't understand You.
Line 20. A tribological test is not a measurement. Use correct terms and correct descriptions of the test process.
Line 21. Degradation - is not a process, that runs on the friction surface during the friction test
In line 35 You changed "practice" to "industry". But what about to make a change in the other places?
The literature review was not changed. It remains chaotic and has minor relevance to Your research. Please, redo it. As it is now, the literature review is unacceptable
Line 67. Once again. How does the high-temperature test refer to Your study? As well as DLC coating (Line 103-110)?
Once again. You did not replace the "pressure material" with the correct term
The sample phase composition is still a speculation. It was not proved by any test. The authors should do the phase identification research, add these results to the manuscript, and then - submit it once again. The science is done in this way, not another. Any of Your words should have firm proof. This time, they are not proven enough.
The structure of the manuscript is still mixed. The results, materials, and research methods are mixed. This makes understanding (together with uncomprehensive language) very hard. I still recommend You follow the recommended article structure. For example, I could not find it, have You included the SEM-sample preparation procedure in the manuscript, or this methodology is still "top-secret"?
The friction test methodology was described incorrectly. The radius - is not a tribological factor. The tribological factors are the type of contact, contact pressure, sliding speed, and lubrication conditions. A change in turning radius changes the sliding speed. So please, describe the test procedure and test results correctly from the tribological point of view. I did not find in the text any comments on my recommendations ti lines 230-235. Do You understand, that changing the sliding radius (when rpm = 250 and test duration is 20 minutes for all cases) changes both sliding speed and sliding distance? Do You understand, that all Your tests were done in dissimilar conditions, and they can not be compared? The wear test should be done once again, in identical speed-distance conditions.
As I said in the previous report, the light microscopy images of the friction surface should be added with SEM images. Yes, light microscopy images may be enough due to Your personal experience, but they are not enough for the reader. You publish the manuscript for the Reader, not for Yourself. Please, add the research, and resubmit the manuscript.
I am sorry, but I can not recommend the manuscript for publication.
Comments on the Quality of English Language
The English style is still poor. I recommend the Authors to do a professional proofreading of the text.
Round 3
Reviewer 1 Report
Comments and Suggestions for Authors
Thank you for the improvements, which answered to almost all my queries. In query 10, regarding figure 7, you say in the paper that "An example of the comparison shown in Figure 7 shows the measured COF curves 265 for the material X153CrMoV12, which were obtained after averaging all the measured results for the given materials." Do not build a curve on average results, because the curve is not real. It is recommended to present a representative curve instead.
Comments on the Quality of English LanguageGood, minor review required
Author Response
Dear Reviewer,
Figure 7 shows the comparison of COF values depending on the measurement methodology for material X153CrMoV12, turning radius 23 mm, load 10 N. This figure remains unchanged because it is the specific values of 1 measurement (for both samples) and is not based on the averaging of 3 values , as we stated originally. An error occurred on our side when describing the image in the text. We originally listed the image correctly with an incorrect description (we wrote: average of 3 measurements). Once again, thank you very much for your reviews, which have taken our article to a higher level.
Reviewer 3 Report
Comments and Suggestions for Authors
Dear Authors, thank You for the work done